# Antioxidant Peptides Derived from Woody Oil Resources: Mechanisms of Redox Protection and Emerging Therapeutic Opportunities

**DOI:** 10.3390/ph18060842

**Published:** 2025-06-04

**Authors:** Jia Tu, Jie Peng, Li Wen, Changzhu Li, Zhihong Xiao, Ying Wu, Zhou Xu, Yuxi Hu, Yan Zhong, Yongjun Miao, Jingjing Xiao, Sisi Liu

**Affiliations:** 1State Key Laboratory of Utilization of Woody Oil Resource, Hunan Academy of Forestry, Changsha 410004, China; tujiashanito@gmail.com (J.T.); 19918130756@163.com (J.P.); xzhh1015@163.com (Z.X.); 17375712512@163.com (Y.H.); 15616334496@163.com (Y.Z.); yjmiao1971@126.com (Y.M.); xjj0806@126.com (J.X.); liusisi274@126.com (S.L.); 2Lipid Molecular Structure and Function Key Laboratory of Hunan Province, Changsha 410004, China; 3Woody Oil Resources Utilization Science Key Laboratory, National Forestry and Grassland Administration, Changsha 410004, China; 4School of Food Science and Bioengineering, Changsha University of Science & Technology, Changsha 410114, China; m18390678979@163.com (Y.W.); xz_jnu@126.com (Z.X.)

**Keywords:** antioxidant peptides, woody oil resources, signal transduction pathways, Keap1/Nrf2/ARE, PI3K/Akt, AMPK, JAK/STAT, bioavailability, functional foods, molecular docking

## Abstract

Antioxidant peptides derived from woody oil resource by-products exhibit strong free radical scavenging abilities and offer potential applications in functional foods, nutraceuticals, and cosmetics. This review summarizes the latest advances in preparation technologies, including enzymatic hydrolysis, microbial fermentation, chemical synthesis, recombinant expression, and molecular imprinting, each with distinct advantages in yield, selectivity, and scalability. The structure–activity relationships of antioxidant peptides are explored with respect to amino acid composition, molecular weight, and 3D conformation, which collectively determine their bioactivity and stability. Additionally, emerging delivery systems—such as nanoliposomes, microencapsulation, and cell-penetrating peptides—are discussed for their role in enhancing peptide stability, absorption, and targeted release. Mechanistic studies reveal that antioxidant peptides from woody oil resources act through network pharmacology, engaging core signaling pathways, including Nrf2/ARE, PI3K/Akt, AMPK, and JAK/STAT, to regulate oxidative stress, mitochondrial health, and inflammation. Preliminary safety data from in vitro, animal, and early clinical studies suggest low toxicity and favorable tolerability. The integration of omics technologies, molecular docking, and bioinformatics is accelerating the mechanism-driven design and functional validation of peptides. In conclusion, antioxidant peptides derived from woody oil resources represent a sustainable, multifunctional, and scalable solution for improving human health and promoting a circular bioeconomy. Future research should focus on structural optimization, delivery enhancement, and clinical validation to facilitate their industrial translation.

## 1. Introduction

Woody oil resources refer to the lipid resources extracted from the seeds or fruits of trees or shrubs, and are widely used in food, cosmetics, industry, and biofuels, among other fields. These vegetable oils are not only favored for their unique nutritional components and health benefits but are also promoted globally due to their sustainability and low environmental impact. Woody oil resources such as olives, palms, coconuts, and tea oil plants (Table 1) not only provide important economic resources for humans but also play a crucial role in maintaining ecological balance and promoting rural economic development.

Woody oil resources are important sources of edible oils or industrial raw materials with high economic value. During the processing of these resources, a large amount of by-products such as oilcake [12], oil residues [13], shells [14], and other remnants are often produced. If these by-products are not fully utilized, it not only constitutes a waste of resources but may also pose environmental issues. Antioxidant peptides from woody oil resources are small molecular peptides produced from the seeds or fruits of woody plants through enzymatic methods and have potent antioxidant properties [15]. These peptides can capture free radicals, slow lipid peroxidation, and protect cells from oxidative damage [16]. Producing antioxidant peptides from woody oil resources not only enhances the added value of the oil but also increases its potential applications in food, cosmetics, and health products, while helping to promote the effective use of agricultural by-products and supporting sustainable agricultural practices [17]. These characteristics make antioxidant peptides from woody oil resources of significant research and commercial value in the health and nutrition sectors.

Although woody oil resources are primarily used for extracting vegetable oils for use in food [18], industry, and biofuels [19], their by-products such as seed meal, which are rich in protein resources, are often not fully utilized, leading to a waste of potential value. Therefore, the high-value processing of by-products from woody oil resources, especially the extraction of active peptides, has attracted widespread attention [20,21,22]. This paper will elaborate in detail on the following aspects: (1) the progress of research into the methods and functions of antioxidant peptides prepared from woody oil resources; (2) the conformational relationship of antioxidant peptides from woody oil resources and their impact on biological activity; (3) the application of antioxidant peptide delivery systems from woody oil resources; and (4) the signaling pathway regulatory mechanism of antioxidant peptides from woody oil resources. The safety, immunogenicity, and toxicity of antioxidant peptides from woody oil resources will also be discussed, as will the clinical translation and ongoing clinical trials of these peptides. Finally, the prospects for the application of antioxidant peptides from woody oil resources and their potential for industrialization will be explored. These studies will help to utilize the by-products of these resources more effectively and promote the use of active peptides in health and disease prevention.

## 2. Advances in the Preparation of and Functional Insights into Antioxidant Peptides Derived from Woody Oil Resources

Antioxidant peptides derived from woody oil can be prepared using various methods, with the appropriate technique selected according to specific needs (Figure 1). Peptides with antioxidant activity can be efficiently released from natural proteins through enzymatic hydrolysis and microbial fermentation, both of which offer mild reaction conditions and high selectivity, making them suitable for use in functional foods, nutraceuticals, and pharmaceuticals. Although chemical hydrolysis is less selective, it is cost-effective, easy to operate, and suitable for large-scale production, and has thus been widely applied in the food, feed, and cosmetic industries. For studies that require precise control of peptide sequences, chemical synthesis, recombinant DNA technology, and computer-aided design (CAD) provide powerful tools that enable the synthesis of peptides of specific sequences, laying the foundation for the study of peptide structure–activity relationships and mechanisms of action. In addition, molecularly imprinted polymer (MIP) post-modification technology, as an emerging technology, is able to selectively extract and enrich the target antioxidant peptides, which opens up a new direction for antioxidant peptide preparation and research. Table 2 lists the advantages and disadvantages of each preparation method of antioxidant peptides. By clarifying the preparation purpose and choosing the most suitable technical means, the target antioxidant peptides can be obtained efficiently to meet the diversified needs in different fields.

### 2.1. Large-Scale Production with Low Purity Requirements

Microbial fermentation produces small-molecule peptides with different biological activities under mild conditions through microbial metabolic activities. This technique has a wide range of applications in the fields of food, pharmaceuticals, nutraceuticals, and cosmetics. In the study of Hassan et al. [31], a mutant strain of *Bacillus licheniformis* was used to synthesize low-cost and potent Bacillus peptides using a deep fermentation technique. Bacillus peptides are often used as additives in animal feed and other biomedical fields. Saubenova et al. [32] in their article describe the ability of Lactic Acid Bacteria (LAB) to produce peptides with antimicrobial, antioxidant, antihypertensive, and antidiabetic properties from different whey proteins using enzymatic digestion and microbial fermentation. These peptides have promising applications as antioxidants and biological preservatives in the pharmaceutical and food industries.

Chemical hydrolysis is a method of obtaining small-molecule peptides by treating proteins with acid or alkali solutions to break their peptide bonds [33]. This method has some applications in the preparation of bioactive peptides, and is the method often used in the initial preparation of soybean peptides. Chemical hydrolysis has the advantages of low cost and simple operation in the preparation of peptides, but it is gradually being replaced by more advanced technologies such as enzymatic methods due to its drastic reaction conditions, unstable product quality, and potential risks to the environment and human health.

Enzymatic preparation of peptides is a technique widely used in food, pharmaceutical, and biotechnology fields. The main advantages are mild reaction conditions, high selectivity, high product purity, and the ability to prepare peptides with specific functions through specific enzymatic sites. It has shown excellent results in the development of peptides with antioxidant, blood pressure-lowering, anti-aging functions, etc. Wang et al. [34] hydrolyzed walnut proteins using trypsin and a combination of enzymes, and found that the resulting walnut peptides could improve Lipopolysaccharide(LPS)-induced memory deficits by modulating inflammatory responses and oxidative stress in the brain. In addition, Liu et al. [35] used neutral enzymes to hydrolyze walnut proteins and obtained peptides with high Angiotensin-Converting Enzyme (ACE) inhibitory activity, suggesting their potential as functional food ingredients with antihypertensive properties.

Overall, the widespread application of these three methods has greatly contributed to the rapid development of peptide-based products across various industries. Each method offers distinct advantages and can complement the others to meet diverse needs and technical requirements.

### 2.2. Synthesis of Functional Peptides with Precise Control

Chemical synthesis is a method used to produce peptides with specific amino acid sequences through chemical reactions. It allows precise control over the sequence and is widely applied in drug discovery, functional food development, and vaccine research. This approach is particularly well-suited for the rapid synthesis of short- and medium-length peptides. Some studies [36] have designed chemically synthesized, modified analogs of human glucagon-like peptide-1 (MGLP-1) and found that the synthetic MGLP-1 is more resistant to trypsin and pancreatin than natural Glucagon-like peptide-1 (GLP-1), suggesting that further research and development of MGLP-1 will increase its potential as a therapeutic agent for GLP-1 analogs in type 2 diabetes patients. While chemical synthesis enables precise peptide production, it has cost-prohibitive scales and technical limitations in long-chain peptides.

With continuous technological advancement, recombinant technology plays an important role in peptide production, especially for long-chain and complex peptides. Through genetic engineering, this technique enables the expression of high-yield, high-purity peptides with specific functions by integrating target peptide genes into host cells [37]. It is commonly used in the production of peptide drugs, vaccines, and functional peptides. Nazarian-Firouzabadi et al. [38] in their article reviewed the recent advances in the design and production of recombinant antimicrobial peptides (AMPs), which are effective in protecting resources from diseases, and as effective plant protection agents, antimicrobial peptides have an important potential for application in plant molecular breeding.

Computer-driven technology (CAD) [39], on the other hand, optimizes peptide sequence design, three-dimensional structure prediction, and bioactivity assessment through computer-aided design [40], bioinformatics analysis, and artificial intelligence technology, greatly improving the efficiency and precision of peptide design. In terms of the multi-objective design of peptides, computer technology is able to simultaneously design peptides with multiple functions, such as antioxidant, anti-inflammatory, and anti-tumor functions, which reduces the experimental cost, enhances the biological activity and stability of peptides, and promotes the wide application of peptide products in the fields of medicine, food, cosmetics, etc. Du et al. [41] used the newly established quantitative constitutive relationship model and the improved hydrolysis simulation tool, R Peptide Cutter, to screen highly active antioxidant dipeptides from sorghum proteins, and found that the improved tools and knowledge can facilitate the research and development of antioxidant peptides. In addition, Medeiros et al. [42] found that computer technology can help to provide important insights into therapeutic strategies for diabetes and provide promising leads for the screening of new therapeutic agents in future trials.

### 2.3. Peptide Preparation with High Selectivity and Precision Recognition

The core advantage of molecularly imprinted polymer (MIP) post-modification technology lies in its ability to mimic natural antibodies by creating polymers with specific recognition sites that can accurately identify and bind target peptides or proteins [43]. Compared to traditional antibodies or other isolation techniques, MIPs are more chemically and thermally stable and can be reused for long periods of time in complex environments, which makes them show unique potential for several applications. Yang et al. [44] used a combination of epitope and surface-restricted imprinting to synthesize a molecularly imprinted polymer (MIP) with high affinity for peptides containing acetylated lysine (Kac). The resulting Kac-MIP was able to distinguish acetylated lysine peptides from their unmodified counterparts and showed strong binding to Kac-peptides with different sequences. This indicates that MIPs have the potential to serve as antibody mimics in post-translational modification (PTM) analysis.

Due to their high specificity, MIPs are often used to extract target bioactive peptides from complex biological samples. For example, MIPs can selectively capture specific peptides during protein extraction, reducing reliance on large volumes of chemical reagents commonly used in traditional purification methods, thereby minimizing environmental impact and lowering costs [45]. In their paper, Incel et al. [46] demonstrate three molecularly imprinted polymer (MIP)-based reagents for the enrichment and subsequent characterization of Protein histidine phosphorylation (p-His) peptides, which demonstrate that molecularly imprinted polymers can enrich p-His peptides under mild pH conditions, which opens up new possibilities for highly robust modification-specific peptide enrichment strategies.

By designing molecularly imprinted polymers with targeted recognition sites, peptides with biological activities such as antioxidant, anti-inflammatory, and anti-tumor can be precisely identified, which provides new ideas for the development of functional foods, drugs, and nutraceuticals; molecular imprinting within double-crosslinked micelles provides a highly efficient method for preparing water-soluble peptide-bound nanoparticle receptors. In the work of Zangiabadi et al. [47], it was found that molecular imprinting of peptides in surface-nucleated double-crosslinked micelles produced hydrophobic pockets complementary to the peptide’s hydrophobic side chains, so that MBAm could be used as a hydrogen-bonded functional crosslinking agent instead of specially designed functional monomers without limiting the selectivity, thus greatly simplifying the preparation of artificial peptide receptors.

## 3. Effects of Structure–Activity Relationship of Antioxidant Peptides

The structure–activity relationship of antioxidant peptides is a complex interplay of amino acid sequence, molecular weight, and three-dimensional structure (Figure 2). Each factor has an impact on the peptide’s interaction with free radicals and other oxidizing substances as well as its stability. With further research, scientists can optimize these structural factors to enhance the antioxidant capacity of antioxidant peptides and expand their applications in industries such as functional foods, nutraceuticals, and even cosmetics.

### 3.1. Effects of Amino Acid Sequence on Activity

In recent years, significant progress has been made in studying the effects of amino acid sequences of antioxidant peptides on oxidative activity [48]. Researchers have identified various amino acids with efficient antioxidant capacity [49] (Figure 3), such as glutamic acid, cysteine [50], histidine, lysine, tryptophan, and tyrosine [51]. These amino acids exhibit antioxidant effects due to the chemical nature of their side chains, which can directly react with free radicals [52].

In their article, Ishak and Sarbon [53] mention that hydrophobic amino acids containing nonpolar fatty groups such as Val, Trp, Tyr, Leu, Ile, His, and Pro react with hydrophobic polyunsaturated fatty acids (PUFAs) and have significant free radical scavenging effects in high-fat foods. Similarly, Kumar et al. [54] added collagen peptides rich in aromatic amino acids and histidine to cookies, which gave the cookies higher protein and antioxidant potentials; 2,2-Diphenyl-1-picrylhydrazyl (DPPH) free radical scavenging activity was increased to 74–88%, the iron reducing power was increased to 0.88–0.91, and the marine collagen peptide (MCP) peptide addition enhanced the health benefits of the cookies. However, it should be noted that the effect of amino acid sequence on antioxidant activity is complex and influenced by many factors, including the overall structure of the protein and environmental conditions. Therefore, the impact of amino acid sequence may vary for different proteins and antioxidant systems. Wu et al. [55], in their study, assessed the antioxidant activity of peptides by replacing redox-related amino acid residues with inactive ones and found that these substitutions did not reduce the peptides’ antioxidant capacity. This implies that the antioxidant activity of peptides is not merely a result of the superposition of active amino acids. The antioxidant activity may also be influenced by other factors, such as spatial effects and the 3D structures affected by intramolecular interactions.

### 3.2. Effects of Molecular Weight on Activity

The optimal molecular weight range for antioxidant peptides is not well defined, but it is generally believed that smaller peptides are more likely to penetrate cell membranes and function inside the cell [56,57], interacting with target free radicals and thus exerting antioxidant effects. Chen et al. [58] developed regular soy milk and its corresponding probiotic yogurt with simulated gastrointestinal digestion and found that the antioxidant activity of the fractions with MW < 10 kDa was about 15% higher than that of the fractions with MW in the range of 10–50 kDa. It has also been demonstrated that lower-molecular-weight Jellyfish collagen Hydrolysate (JCH) is more protective against photoaging in mouse skin, so JCH-rich diets and nutrients may help to minimize UV damage to the skin. This finding suggests that JCH may serve as a novel natural anti-photoaging agent [59]. Similarly, a study [60] that extracted peptides from olive oil residue and assayed them for antioxidant activity found that peptides with molecular weights less than 3 kDa showed better antioxidant capacity than those with higher molecular weights. Low-molecular-weight peptides, due to their small molecular weight, are able to enter directly into cells, participate in metabolism, exert antioxidant effects [61], and improve bioavailability. These characteristics make low-molecular-weight peptides have a wide range of application prospects in the field of food, nutraceuticals, and pharmaceuticals.

### 3.3. Effects of Three-Dimensional Structure on Activity

The three-dimensional structure of a peptide determines its contact area and binding mode with free radicals, thus affecting its ability to scavenge them. Secondary structures such as α-helix, β-sheet, β-turn, and random coil influence the way the peptide binds to free radicals or metal ions, thereby impacting its antioxidant capacity. For instance, Martí-Quijal [62] in his study found that the secondary structure of fish head peptides could be altered by the pulsed electric field (PEF) treatment technique to improve their antioxidant capacity, increasing the diversity of peptides in the fish head, and a wider range of peptides were found in the pre-treated samples compared to the control samples. Moreover, Zhang et al. [63] also found that the peptides PGMLGGSPPGLGGSPP and SDGSNIHFPN identified in snakehead soup, which have β-turn, β-sheet, or α-helix structures, exhibited relatively high antioxidant activity, while the DPPH radical scavenging activity of the randomly coiled SVSIRADGGEGEVTVFT was relatively low. The effects of three-dimensional structures on antioxidant activity are significant and multifaceted. Through molecular mechanics, quantum chemistry, and machine learning, the relationship between 3D structure and antioxidant activity can be deeply understood and provide important theoretical support for new drug development and natural product research.

## 4. Delivery Systems with Antioxidant Peptides

Antioxidant peptides typically have a short half-life and are susceptible to degradation by stomach acid, pepsin, or intestinal proteases [64]. Therefore, direct oral administration or other traditional drug delivery methods often fail to effectively deliver bioactive peptides, resulting in reduced biological activity and therapeutic efficacy. Consequently, researching and developing new delivery systems is crucial for enhancing the stability, targeting, absorption efficiency, and bioavailability of antioxidant peptides. This section primarily discusses how to improve the bioavailability, stability, and targeted delivery efficiency of peptides within the body, covering the technologies and methodologies involved (Figure 4).

### 4.1. Nanotechnology Applications

To improve the absorption and transport of bioactive peptides, nanotechnology applications focus on enhancing bioavailability, stability, targeted transport, and controlled release. Common nanotechnologies include nanoparticles (NPs) and nanoliposomes.

Nanoparticles [65] are tiny particles composed of materials such as metals, polymers, or lipids. Bioactive peptides can be bound to nanoparticles by adsorption, coating, or covalent linkage to enhance selective recognition and delivery to target cells through targeted modification. Yang et al. [66] prepared chitosan nanoparticles to encapsulate hydrophilic exenatide and further coated them with sodium alginate, and they found that the release of chitosan/alginate nanoparticles (CS-TPP-ALG) in simulated intestinal fluid (SIF) reached 90% within 1 h, whereas the release in simulated gastric fluid (SGF) was below 10% within 3 h. This indicated that sodium alginate enteric coating could protect chitosan nanoparticles from gastric acid damage and thus effectively controlled release, and the oral bioavailability was also calculated to be about 9.16%, showing great potential for oral administration of the antidiabetic peptide drug exenatide. Jeong et al. [67] combined peptides with NPs after functionalizing the peptides as targeting agents and found that in addition to having the ability to encapsulate and protect the therapeutic agent, the NPs could be engineered to selectively deliver the drug to the target tissues, and that different targeting peptides coupled to different types of NPs could provide a more effective and adaptable drug delivery system.

Another commonly used nanotechnology is liposomes. Liposomes [68] are tiny droplets composed of phospholipids and cholesterol that can encapsulate biologically active peptides to form nanoscale delivery systems. Liposomes can enhance the stability of peptides and promote their binding and permeation across cell membranes, thus enhancing peptide bioactivity. Liposomes and nanoliposomes protect peptides while enabling targeted delivery by altering their surface properties. Wang et al. [69] prepared four precursor liposomes loaded with walnut peptides, and they found that the DPPH radical scavenging rate of the four liposomal progenitors loaded with walnut peptides was maintained at about 60.0% after 25 days of storage. In addition, the minimum inhibitory concentration (MIC) values of these four liposomes against *E. coli* and *S. aureus* were 0.7 and 0.6 g/mL, 0.6 and 0.4 g/mL, 0.5 and 0.5 g/mL, and 0.3 and 0.4 g/mL, respectively. The experimental results indicated that walnut peptides encapsulated in liposomes possessed higher antioxidant activity and resistance to environmental stresses, and all the walnut peptides retained their antimicrobial activity. Similarly, Zhang et al. [70] found that encapsulating walnut peptides in nanoliposomes could improve the storage and in vitro digestive stability of walnut peptides, and the unprotected walnut peptide fractions were released up to 90.0% within 30 min in gastrointestinal (GI) digestion, while encapsulated walnut peptides were released by about 35.2% and 66.1% in 30 min in SGF and SIF digestive systems, respectively, indicating that the nanoliposomes were effective in protecting the acidic or alkaline environment in the GI system. After intestinal digestion, the nanoliposomes released about 83.3% of the peptides, and these results indicated that the nanoliposomes could provide sufficient protection for walnut peptides during storage and digestion. However, liposomes, as a commonly used carrier in peptide delivery, have some limitations [68]. The peptide delivery efficiency of liposomes may be affected by various factors, such as the nature of the loaded peptide, the composition of the liposome, and the method of preparation, making it difficult to ensure their effectiveness in different situations.

### 4.2. Biological Polymer Carriers: Natural Polymers and Synthetic Polymers

Biopolymer carriers can improve the stability and bioavailability of antioxidant peptides and further enhance their targeting and controlled release properties through modification and functionalization. A common biopolymer carrier is poly lactic-*co*-glycolic acid (PLGA).

PLGA is a synthetic biodegradable polymer commonly used in the preparation of particles or nanoparticles [71]. It exhibits excellent controlled release properties and protects antioxidant peptides from degradation in the gastrointestinal environment, enhancing their oral bioavailability. Ismail et al. [72] prepared liraglutide-loaded PLGA nanoparticles (PLGA NPs) using a double emulsion solvent evaporation method and found that the PLGA NPs protected the encapsulated liraglutide from a simulated gastric environment. They also performed cellular modeling and found that liraglutide encapsulated in PLGA NPs had a 1.5-fold higher permeability in Caco-2 cells compared to liraglutide solution, indicating that PLGA NPs have great potential as a vehicle for the delivery of oral GLP-1 analogs by protecting liraglutide from enzymatic degradation and enhancing its permeability across the intestinal epithelium. However, PLGA as a delivery system still faces many challenges; for example, PLGA has issues with insufficient drug-carrying efficiency [64].

### 4.3. Targeted Delivery Systems

In the area of antioxidant peptide uptake and transportation, the application of targeted delivery systems can significantly improve their efficiency and effectiveness. Targeted delivery systems [73] are designed to deliver antioxidant peptides directly to specific cells, tissues, or organs to maximize their bioactivity and reduce side effects. Common technologies used for the targeted delivery of bioactive peptides include cell-penetrating peptide technology and smart response delivery systems.

Cell-penetrating peptide (CPP) technology [74] is used to enhance the penetration of biomolecules (e.g., drugs, proteins, nucleic acids, etc.) across the cell membrane, allowing them to enter the cell interior. This technology typically involves the design and synthesis of specific peptide sequences that interact with specific receptors or lipids on the cell membrane to facilitate the passage of biomolecules through the cell membrane. Cell-penetrating peptide technology has a wide range of applications, especially in the fields of drug delivery [75] and gene transfection. It can overcome the barrier effect of the cell membrane and effectively deliver drugs or genetic materials to the inside of target cells, thus improving therapeutic efficacy or research efficiency.

Dougherty et al. [76] were able to make pinned peptides highly cell-permeable by concatenating cyclic cell-penetrating peptides to their N-terminal, C-terminal, or pinned units. The resulting cyclic CPP pinned peptide concatenates can be used to target previously challenging proteins. Ildefonso et al. [77] described a novel method of intraretinal delivery of small peptides using an adeno-associated virus (AAV) vector that binds the Nrf2 derivative peptide of Keap1. The sequence of the Nrf2 peptide was fused to a cell-penetrating peptide sequence (TatNrf2mer), and the results showed that in eyes treated with the sGFP-TatNrf2mer vector, the level of the nitro-tyrosine-modified proteins were reduced by more than 50%, suggesting that the TatNrf2mer peptide protects the retina from oxidative stress by modulating the Nrf2 signaling pathway in the retina. However, cell-penetrating peptides also have shortcomings. For example, some cell-penetrating peptides may lack sufficient specificity, resulting in them acting not only on the target cell but also on other cells.

Smart response delivery systems are delivery systems [78] that can sense changes in the external environment and autonomously adjust drug release in response to these changes. Smart nanoparticles have constituted an excellent platform for achieving effective cancer therapy by responding to endogenous (pH, enzyme, or redox gradients) or exogenous stimuli (light, temperature, ultrasound, magnetic, and electric fields) released specifically at the tumor site [79]. Ma et al. [80] designed and developed pH-responsive benzimidazole–chitosan quaternary ammonium salt (BIMIXHAC) nanogels for the controlled release of doxorubicin hydrochloride (DOX), and they found that the antioxidant capacities were 75.05%, 72.75% and 68.45% for BIMIXHAC-HA, BIMIXHAC-SA, and BIMIXHAC-CMC, respectively. The antioxidant capacity of BIMIXHAC-HA (80.35%), BIMIXHAC-SA (77.39%), and BIMIXHAC-CMC (70.73%) was found to be 75.05%, 72.75%, and 68.45%, respectively, and the scavenging capacity of superoxide anion radical was found to be BIMIXHAC-HA (80.35%), BIMIXHAC-SA (77.39%), and BIMIXHAC-CMC (70.73%). The proposed pH-responsive DOX-loaded nanogels based on BIMIXHAC nanogels as nanocarriers for anticancer drugs provide a promising controlled drug delivery strategy, enhancing antioxidant and anticancer effects.

Although smart delivery systems are designed to achieve targeted release, in practice, targeting effects may be affected by biodiversity and individual differences, resulting in unstable or inconsistent effects. Additionally, the preparation process of certain smart response delivery systems is complex and requires high-precision technology and equipment support, which is costly and technically demanding.

### 4.4. Encapsulated Delivery Systems for Microcapsules and Nano-Emulsions

After oral administration, antioxidant peptides first enter the stomach, where gastric acid (low-pH environment) and pepsin disrupt the structure of the peptide, resulting in the loss of its biological activity. Scientists have developed a variety of delivery systems such as microencapsulation techniques and nano-emulsions.

Microencapsulation technology involves using natural or synthetic polysaccharides, proteins, or lipid materials to encapsulate antioxidant peptides in microcapsules. This forms a protective barrier that prevents degradation in the stomach, facilitates release and absorption in the small intestine, prevents volatilization or reactions, and masks undesirable odors, among other purposes. A study [81] was conducted to prepare microcapsules with chitosan and caffeic acid-grafted chitosan (CA-g-Ch) as carriers for efficient encapsulation of bioactive peptides from the pupa of the silkworm, Bombyx mori, using microfluidic technology. The encapsulation efficiency of CA-g-Ch–sodium alginate microcapsules was increased by 96.61 ± 1.95%, and the scavenging activity of DPPH free radicals was increased by 31.15 ± 0.99%. In the gastrointestinal digestion test, the release of peptides from the CA-g-Ch–sodium alginate microcapsule group was 9.76 ± 0.23%, which was 5% lower compared to the control group. The microcapsules prepared by the new process of this study have improved thermal stability and gastric acid resistance, and are a new cost-effective loaded hydrophilic drug carrier, expanding the utilization of API products.

Nano-emulsion technology is a method of preparing emulsions using nanotechnology. Nano-emulsions can be categorized into oil-in-water (O/W) and water-in-oil (W/O) types. Nano-emulsions are capable of dissolving antioxidant peptides in the oil phase and increasing the surface area of the peptides through nano-sized oil droplets, thus enhancing their absorption [82]. Nano-emulsion delivery systems hold great promise in the food, pharmaceutical, and cosmetic fields by enhancing the bioavailability of hydrophobic ingredients and achieving precise targeting and smart release. Fardous et al. [83] developed an organic gel-based nano-emulsion designed to efficiently deliver hydrophobic drugs by encapsulating them in organic gel droplets that can be gelatinized in situ. The nano-emulsions prepared in this study could effectively encapsulate and deliver hydrophobic drugs, and the physicochemical properties of hydrogel nano-emulsions (G/W NE) were unaffected by storage conditions and remained stable over a period of 6 months. In addition, G/W NE did not affect the mitochondrial and metabolic activities of primary rat hepatocytes and was biocompatible. Therefore, G/W NE is a promising hydrophobic drug carrier to improve the efficiency of drug delivery in vivo. Nano-emulsion encapsulation is a promising technology for natural antioxidants, as it not only improves the physicochemical stability of natural antioxidants but also prevents their deterioration during food processing [84].

## 5. Studies on the Signaling Pathway Regulation Mechanism of Antioxidant Peptides from Woody Oil Resources

Antioxidant peptides derived from woody oil resources exhibit unique network effects in the regulation of biological activities that do not depend on the linear transmission of a single signaling pathway (Figure 5), but rather on the formation of sophisticated synergistic effects in gene expression, redox homeostasis, and cell fate regulation through a multidimensional, dynamically interconnected signaling network. The core of this multi-pathway linking mechanism lies in the ability of antioxidant peptides to target multiple key signaling nodes and trigger hierarchical, time-sequential regulatory responses through molecular interactions, metabolic remodeling, and cross-system communication. When the antioxidant peptide enters the cell, the active moiety in its molecular structure can bind directly to the Keap1 protein to release its inhibitory effect on Nrf2, causing Nrf2 to enter the nucleus and activate the ARE element to drive the expression of antioxidant enzymes such as heme oxygenase-1 (HO-1) and glutathione synthetase (GCL) [85]; at the same time, the peptide can enhance cell survival signaling by activating the PI3K/Akt pathway and inhibiting the activity of the pro-apoptotic protein Bad [86], further enhancing the antioxidant effect through Akt-mediated Nrf2 phosphorylation. In particular, the derivatives produced by antioxidant peptides during metabolism could significantly alter the intracellular AMP/ATP ratio and activate the energy sensor AMPK, thereby enhancing mitochondrial function and superoxide dismutase (SOD) activity, forming a positive metabolic–antioxidant cycle. This mode of modulation from molecular target to systemic network enables woody oil resource antioxidant peptides to overcome the limitations of traditional single-target interventions and realize “multi-target correction” in oxidative stress-related diseases (e.g., metabolic syndrome and neurodegenerative diseases)—we can enhance endogenous antioxidant defenses via Nrf2/ARE, maintain the cell survival threshold via PI3K/Akt, and balance the metabolic–inflammatory microenvironment via the AMPK-JAK/STAT network [87]. In the future, we need to further analyze the molecular basis of the conformational relationship and pathway selectivity of the peptides, develop a delivery system based on gut flora–host interactions, and establish an integrated multi-omics analysis platform, in order to maximize the advantages of their “network pharmacology” in precision nutritional interventions and cross-scalar disease therapies.

### 5.1. Integrated Regulatory Mechanisms of Core Antioxidant Signaling Pathways

#### 5.1.1. Keap1/Nrf2/ARE Pathway: A Redox-Sensing Hub for Peptide Intervention

The Keap1/Nrf2/ARE pathway is a well-characterized oxidative stress response mechanism that is highly responsive to antioxidant peptides derived from woody oil resources. These peptides can bind to Keap1, releasing its inhibitory effect on Nrf2, which then translocates to the nucleus and activates ARE elements to promote the expression of antioxidant enzymes such as HO-1 and NQO1.

For instance, Zhong et al. [37] identified four functional peptides (e.g., PCRGVLLR and KVLPVPQKA) from walnut proteins fermented by Lactobacillus casei, which showed strong binding affinities to Keap1 (−7.4 to −8.3 kcal/mol) and significantly enhanced antioxidant capacity in vivo. Similarly, Qi et al. [88] reported that the walnut-derived peptide LPLLR inhibited Keap1–Nrf2 interaction through steric hindrance, resulting in increased SOD and CAT activity and decreased MDA levels in H_2_O_2_-exposed Caco-2 cells, indicating its structural targeting potential.

#### 5.1.2. PI3K/Akt Pathway: Enhancing Redox Signaling and Cytoprotection

The PI3K/Akt pathway is a key intracellular signaling cascade that bridges oxidative stress regulation and cell survival. It has been shown that antioxidant peptides derived from woody oil resources can activate this pathway to indirectly enhance Nrf2-mediated transcription of antioxidant genes.

Several studies demonstrated that walnut-derived peptides restore PI3K/Akt signaling under oxidative damage conditions (e.g., H_2_O_2_ exposure), inhibit apoptosis-related proteins, and promote antioxidant enzyme activity [89,90].

#### 5.1.3. JAK/STAT Pathway: Coordinating Anti-Inflammatory and Redox Responses

The JAK/STAT pathway plays a dual role in immune regulation and oxidative stress defense. Recent studies have shown that certain antioxidant peptides derived from woody oil resources can inhibit the IL-6/JAK2/STAT3 axis to reduce neuroinflammation and oxidative damage.

For example, Zhang et al. [91] found that the heptapeptide WCPFSRSF attenuated cognitive impairment induced by sleep deprivation, partially by inhibiting microglial activation and suppressing the JAK/STAT signaling pathway. Additional studies using in vitro models have demonstrated that these peptides can lower ROS levels, upregulate endogenous antioxidant enzymes, and downregulate pro-inflammatory cytokines, achieving a dual anti-inflammatory and antioxidant effect.

### 5.2. Exploration of Auxiliary Pathways and Systemic Crossover Mechanisms

#### 5.2.1. AMPK Pathway: Energy–Redox Coupling and Mitochondrial Repair

AMPK (AMP-activated protein kinase) serves as a central regulator of cellular responses to energy stress (e.g., elevated AMP/ATP ratio). Upon activation, it inhibits mTOR signaling, initiates autophagy, and promotes mitochondrial biogenesis via the SIRT1/PGC-1α axis—thereby restoring both energy and redox balance [92].

Recent studies have highlighted AMPK as a critical node in the action of various antioxidant peptides. For instance, the dietary peptide QEPV was shown to enhance antioxidant enzyme activity and suppress lipid peroxidation in RAW 264.7 cells via activation of the AMPK and PPARα pathways [93]. Similarly, the peptide DDWENWAK from perch hydrolysate improved mitochondrial membrane potential, structure, and ATP production in H_2_O_2_-stressed Caco-2 cells by activating both the AMPK and Nrf2 pathways [94]. These findings support the potential of natural protein-derived peptides to modulate AMPK signaling for energy metabolism and oxidative stress protection.

If antioxidant peptides derived from woody oil resources can be further validated to activate AMPK and enhance mitochondrial repair, they could serve as promising agents for countering metabolic stress in functional food applications.

#### 5.2.2. Emerging Mechanisms: Non-Coding RNAs and Microecological Interactions for Regulation

Beyond classical signaling pathways, non-coding RNAs (e.g., microRNAs) and gut microbiota have emerged as key players in shaping the bioactivity of antioxidant peptides, forming a complex peptide–epigenetic–microecological regulatory axis. Yan et al. [95] reported that the antimicrobial peptide YD downregulated *miR-155* expression, which relieved suppression of *CASP12* and inhibited the NF-κB pathway—ultimately reducing inflammation and oxidative stress. This indicates that peptides can modulate miRNA networks to control immune and oxidative responses. Similarly, *miR-27a* has been shown to regulate the PPARγ–PI3K/Akt–GLUT4 axis, thereby improving insulin resistance in obese mice and 3T3-L1 adipocytes [96].

Peptides influencing miRNA expression could therefore fine-tune cellular metabolism and cytoprotection.

In parallel, peptide–microbiota interactions are gaining attention. High collagen peptide intake has been shown to reshape gut microbial composition and increase the production of short-chain fatty acids (SCFAs) [97]. SCFAs such as acetate, propionate, and butyrate can activate AMPK/PPARγ signaling and suppress c-Jun phosphorylation, thereby inhibiting hepatic stellate cell (LX2) activation and exerting anti-fibrotic effects [98]

While current studies have not directly demonstrated the involvement of antioxidant peptides derived from woody oil resources in miRNA or microbiota regulation, their unique amino acid profiles suggest strong potential in these emerging mechanisms. Future research incorporating the gut–brain axis, miRNA expression profiling, and metabolomics could further reveal their systemic antioxidant and anti-inflammatory capabilities.

## 6. Safety, Immunogenicity, and Toxicity of Antioxidant Peptides Derived from Woody Oil Resources

To assess the biosafety of antioxidant peptides derived from woody oil resources, various in vitro, animal, and human studies have been conducted. These studies evaluate cytotoxicity, metabolic tolerance, and systemic safety across multiple models. Table 3 summarizes key findings from recent research, including acute toxicity assessments, sub-chronic dosing studies, and early-stage clinical trials.

### 6.1. Acute and Sub-Chronic Toxicity

Current rodent data place the no-observed-adverse-effect level (NOAEL) for walnut-derived oligopeptides above 440 mg kg^−1^ day^−1^ for 30 days [102,103], while cell assays show Camellia peptides are non-toxic at ≥200 µg mL^−1^ [99,104]. Longer 90-day studies meeting OECD 408 guidelines are still needed for Camellia and other antioxidant peptides derived from woody oil resources.

### 6.2. Immunogenicity

Antioxidant peptides derived from woody oil resources are not listed among common food allergens, yet novel linear epitopes could arise after hydrolysis or recombinant expression [105]. In silico screens using AllerTOP v2.0 predict low allergenic potential for reported walnut and Camellia peptide sequences [106,107]; nevertheless, basophil activation or human-sera IgE panels are recommended before market entry.

### 6.3. Pro-Oxidant and Metabolic Liabilities

At very high concentrations, sulfur- or aromatic-rich peptides may redox cycle with transition metals [108]; controlled dose–response curves are therefore essential [109]. Peptides <1 kDa are rapidly cleaved and cleared, whereas medium-length sequences (>10 aa) warrant renal clearance profiling in vulnerable populations [110].

## 7. Clinical Translation and Ongoing Human Studies

Despite growing preclinical evidence, the clinical translation of antioxidant peptides derived from woody oil resources remains in its early stages. Human studies to date have primarily focused on safety and exploratory functional outcomes in small cohorts. These early findings provide a foundation for broader clinical application in functional foods, cognitive enhancement, and cosmeceutical formulations. Table 4 provides a summary of currently available human studies, including both completed trials and registered exploratory investigations.

## 8. Application Prospects and Industrialization Potential

Pathway-specific antioxidant peptides derived from woody oil resources demonstrate promising applications across health-related industries. By selectively activating key signaling pathways such as Nrf2/ARE, PI3K/Akt, and AMPK, these peptides enhance cellular antioxidant defenses and suppress chronic inflammation, especially relevant for aging populations and those with metabolic disorders.

(1)Precision Nutrition for Aging and Metabolic Syndromes

Food-derived peptides [112] may have greater stability, safety, absorption efficiency, and biological activity, properties that give them greater potential to alleviate diseases associated with oxidative damage than other antioxidant drugs or phytochemicals. Tonolo et al. [113] synthesized four peptides from milk protein sequences and confirmed their antioxidant effects through Nrf2 nuclear translocation in Caco-2 cells. Among them, KVLPVPEK showed the strongest activation of antioxidant gene expression via the Keap1-Nrf2 pathway.

Yang et al. [114] demonstrated that BSP1 (KKWNP), a peptide from black soybean, reduced lead-induced neuroinflammation by inhibiting NF-κB phosphorylation and suppressing inflammatory cytokines such as IL-1β, TNF-α, NLRP3, and IL-18. Notably, BSP1 also activated the AMPK/SIRT1 pathway by promoting *LKB1* and AMPK phosphorylation, offering neuroprotective potential against Alzheimer’s disease.

For metabolic diseases like type 2 diabetes, peptides with α-glucosidase inhibition or glycogen synthesis regulation are being explored [115]. Jiang et al. [116] showed that Vglycin, a 37-residue peptide isolated from pea seeds, improved insulin sensitivity via PI3K/Akt-mediated phosphorylation of GSK3α/β and upregulation of GLUT4 expression, suggesting its potential in glycemic control.

(2)Functional Foods via Multi-Pathway Synergy

To amplify biological effects, antioxidant peptides can be complexed with phenolic compounds, forming peptide–phenol conjugates that exploit synergistic multi-pathway activation. Choi et al. [117] demonstrated that coupling the oyster peptide QHGV with phenolic acids such as caffeic, gallic, and ferulic acids enhanced its antioxidant activity and suppressed LPS-induced inflammation in macrophages. These conjugates inhibited ROS production and the expression of iNOS and COX-2, while downregulating the phosphorylation of JNK, ERK, and p38 MAPK, highlighting their potential as next-generation functional food ingredients.

(3)Cosmeceutical Applications: Anti-UV and Skin Barrier Repair

Topical application of antioxidant peptides holds potential in anti-photoaging skincare. Cheng et al. [118] found that three yeast-derived fermentation actives could alleviate H_2_O_2_^−^ and UVA-induced oxidative damage in human fibroblasts by activating the PI3K/Akt pathway.

In parallel, Song et al. [119] showed that fermented oat bran extract downregulated JAK/STAT signaling, suppressed inflammatory cytokine transcription, and improved skin barrier function in UVB-damaged keratinocytes. These results support the development of anti-UV serums and barrier-repairing formulations based on antioxidant peptides.

(4)Mechanism-Guided Peptide Innovation for Industrial Translation

With advancements in omics technologies, molecular docking, and pathway-targeted validation, peptide development is shifting from empirical screening to mechanism-based innovation. This transition supports the industrialization of antioxidant peptides derived from woody oil resources antioxidant peptides as high-value bioactives in nutrition, functional foods, and cosmeceuticals.

## 9. Future Perspectives

Antioxidant peptides derived from woody oil resources (e.g., Camellia and walnut) show strong promise as natural agents for oxidative stress management, but several challenges remain before large-scale application can be achieved.

(1)Structure–function elucidation

Future studies should focus on clarifying the structure–activity relationship (SAR) of key peptide sequences using tools like molecular docking, quantitative structure–activity relationship (QSAR) modeling, and AI-driven prediction [106].

(2)Improved bioavailability

Developing effective delivery systems (e.g., nano-emulsions and enteric coatings) is essential to protect peptides from gastrointestinal degradation and enhance systemic absorption [120].

(3)Mechanistic integration via multi-omics

Combining transcriptomics, proteomics, and microbiomics will help unravel how these peptides influence redox pathways, gut–brain signaling, and host metabolism [121].

(4)Human validation

Though walnut peptides have entered clinical trials [122], more human studies are needed for other woody oil resources to assess safety, efficacy, and long-term effects.

(5)Sustainable industrialization

Utilizing seed cake proteins from oil production to generate high-value peptides supports circular bioeconomy goals and clean-label product development [123].

## 10. Conclusions

Antioxidant peptides derived from woody oil resources represent a valuable and underexplored class of bioactives with multifunctional health benefits. This review systematically elucidated their preparation technologies, structure–activity relationships, delivery systems, and molecular mechanisms of action through core and auxiliary signaling pathways—including Nrf2/ARE, PI3K/Akt, JAK/STAT, and AMPK—to regulate redox balance, inflammation, and mitochondrial health, offering promising applications in precision nutrition, functional foods, and cosmeceuticals. Their low toxicity, multi-target actions, and compatibility with food and cosmetic systems support their potential as functional ingredients. Future research should focus on clarifying structure–function relationships, improving oral bioavailability through advanced delivery systems, and validating efficacy in human trials. Leveraging computational tools and omics technologies will accelerate the development of targeted peptide products, enabling the sustainable valorization of woody oil by-products into high-value solutions for nutrition, health, and disease prevention.

## Figures and Tables

**Figure 1 pharmaceuticals-18-00842-f001:**
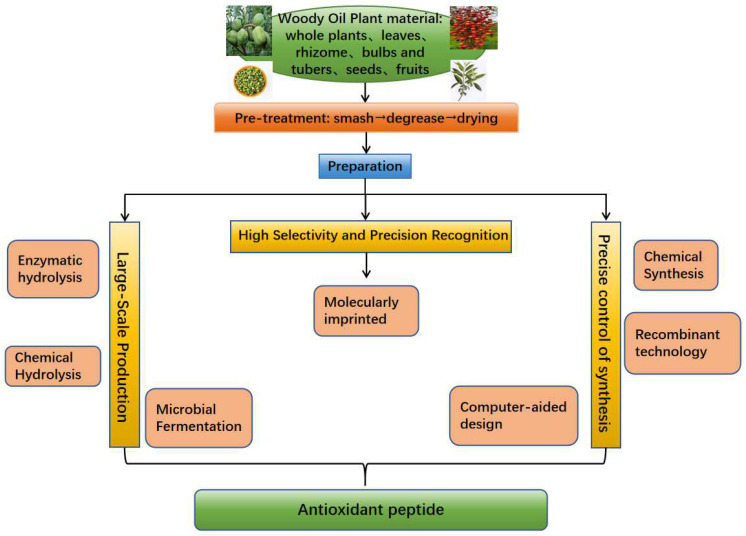
Schematic route from woody oil resources to antioxidant peptides. Pre-treated woody oil seeds or by-products are subjected to either large-scale enzymatic hydrolysis/fermentation or precision synthesis approaches such as chemical or recombinant methods. A subsequent molecular imprinting enrichment step selectively isolates target peptides, which are ultimately applied in food, nutraceutical, and pharmaceutical products.

**Figure 2 pharmaceuticals-18-00842-f002:**
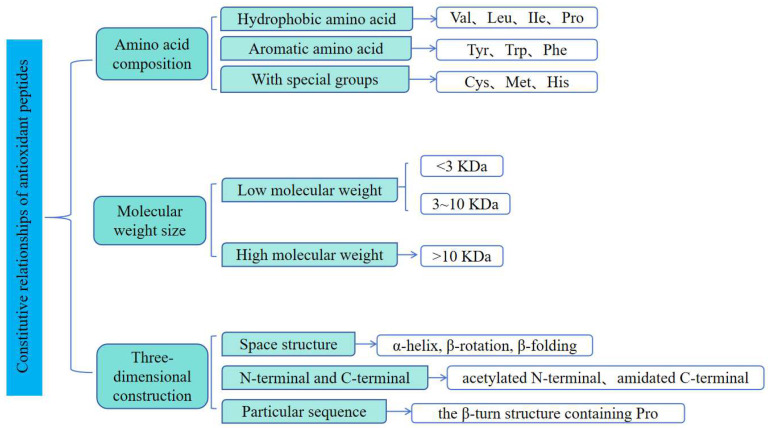
The antioxidant potency of bioactive peptides is dictated by three interlinked factors: amino acid composition, molecular weight range, and three-dimensional conformation.

**Figure 3 pharmaceuticals-18-00842-f003:**
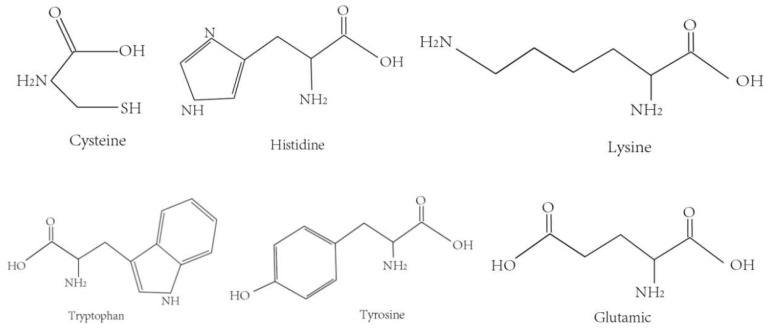
Chemical structures of antioxidant-relevant amino acids (cysteine, histidine, lysine, tryptophan, tyrosine, and glutamic acid), whose thiol, imidazole, ε-amino, indole, phenolic, and γ-carboxylate side chains contribute to radical scavenging and metal-chelating activity in bioactive peptides.

**Figure 4 pharmaceuticals-18-00842-f004:**
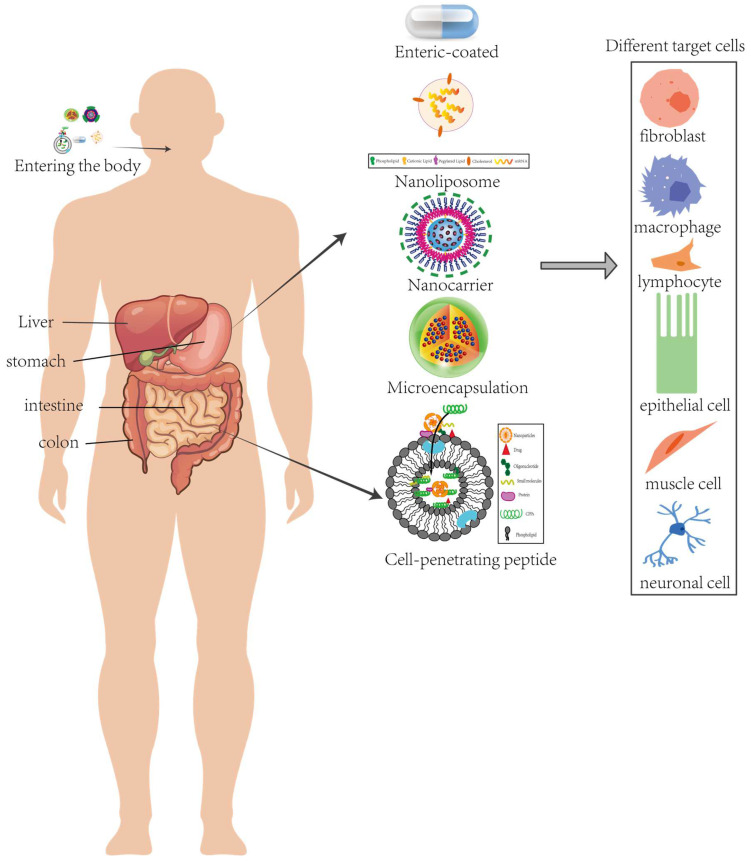
Schematic of oral delivery showing gastrointestinal barriers and five carrier strategies: enteric coating, nanoliposomes, nanocarriers, microencapsulation, and cell-penetrating peptides, which safeguard antioxidant peptides en route to systemic circulation and diverse target cells.

**Figure 5 pharmaceuticals-18-00842-f005:**
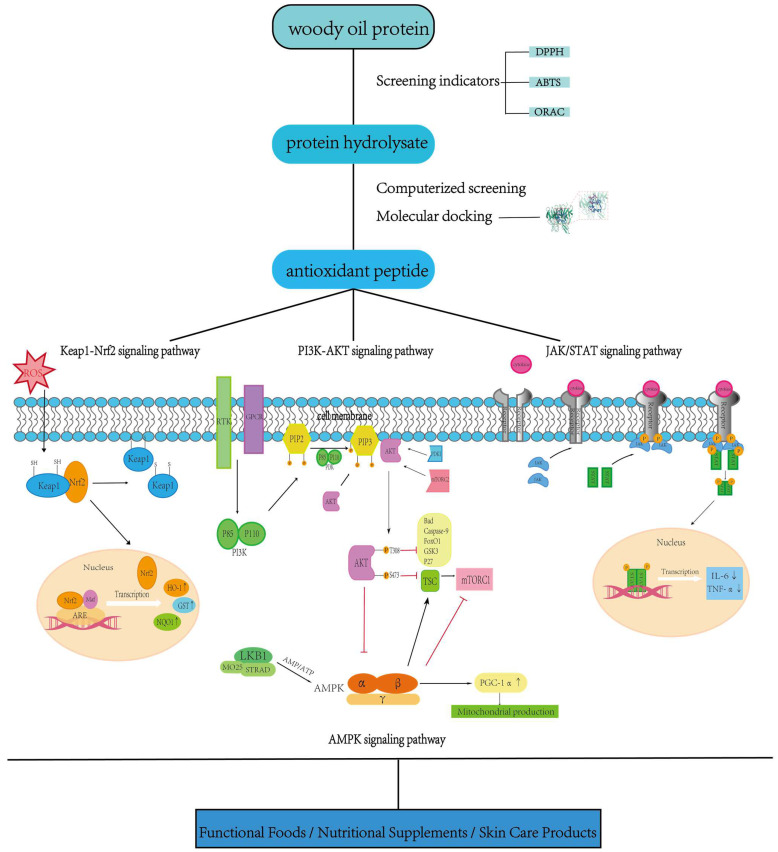
Antioxidant peptides derived from woody oil resources act via network pharmacology, synchronously engaging Keap1–Nrf2, PI3K–Akt, AMPK, and JAK/STAT pathways to coordinate antioxidant gene expression, metabolic–redox homeostasis, and cell survival control. This highlights their multi-target potential for precision nutrition and disease intervention.

**Table 1 pharmaceuticals-18-00842-t001:** Woody oil resources and their products, distribution, applications.

Woody Oil Resources	Main Products	Major Distribution Regions	Application Industries
Olive [1]	Olive Oil	Mediterranean region, Americas, Asia	Food and Cosmetics
Oil Palm [2]	Palm Oil and Palm Kernel Oil	Asia, Africa, Americas	Food, Cosmetics, and Biofuel
Coconut [3]	Coconut Oil	Africa, Latin America, Asia	Food, Cosmetics, and Pharmaceutical Industry
*Camellia oleifera* [4]	*Camellia* Oil	Guangdong, Hong Kong, Guangxi, Hunan, Jiangxi	Edible and Industrial Uses
Walnut [5]	Walnut Oil	China, USA, Turkey, Mexico, Iran	Edible and Medicinal
Almond [6]	Almond Oil	USA, Turkey, Australia, EU, China	Food and Cosmetics
Peony Seed [7]	Peony Seed Oil	Japan, France, UK, USA, China	Food and Cosmetics
Safflower Seed [8]	Safflower Seed Oil	India, Mexico, China	Food and Industrial Applications
Grape Seed [9]	Grape Seed Oil	Europe, Asia, Americas	Food and Cosmetics
*Litsea cubeba* [10]	*Litsea cubeba* Essential Oil	Eastern Asia, Oceania, Pacific Islands	Edible and Medicinal
*Cornus wilsoniana Wangerin* [11]	*Cornus wilsoniana Wangerin* Essential Oil	China	Food and Biofuel

**Table 2 pharmaceuticals-18-00842-t002:** Advantages and limitations of different preparation methods for antioxidant peptides.

Method	Key Advantages	Key Limitations
Chemical Synthesis [23]	High accuracy Good controllability Diversity	Many side reactions High cost Difficult to synthesize long-chain peptides
Chemical Hydrolysis	Rapid reaction Low cost [24] Simple operation	Poor specificity Environmental pollution [25] Amino acid damage
Enzymatic Hydrolysis [24]	Gentle conditions Controlled reaction Safety	Low yield Purification difficulties
Microbial Fermentation [26]	Gentle conditions Eco-friendly Low cost	Limited yield Time-consuming [27] Fermentation conditions require precise control
Recombinant technology [28]	Strong expression orientation Safety Low cost	Purification difficulties Not suitable for small-molecule peptides Long cycle times
Computer-aided [29]	Low cost Efficiency Flexibility	High computer resource requirements [30] Experimental verification needed

**Table 3 pharmaceuticals-18-00842-t003:** Safety modeling of antioxidant peptides from woody oil resources.

Evidence Tier	Model/Protocol	Dose and Duration	Key Observations
In vitro	Human L-02 hepatocytes, alcohol–injury model	Camellia vietnamensis peptide A1–2, ≤200 µg mL^−1^, 24 h	No cytotoxicity to normal cells; restored viability after EtOH insult [99]
Sub-acute in vivo	BALB/c mice, 30-day oral study	Walnut oligopeptides (WOPs) 110–440 mg kg^−1^ day^−1^	No mortality, normal serum liver–renal panels; behavioral indices unchanged [100]
Human (Phase I/II)	RCT, teenagers and elderly (*n* = 36)	WOPs 170 mg or 340 mg day^−1^, 90 days	No adverse events recorded; hematology and biochemistry within reference ranges [101]

**Table 4 pharmaceuticals-18-00842-t004:** Clinical translation of antioxidant peptides derived from woody oil resources.

Evidence Tier	Woody Oil Source and Study ID	Population/Design	Principal Findings
Development stage	Walnut oligopeptides (ChiCTR1900028160)—90-day, randomized, double-blind, placebo-controlled trial	18 teenagers + 18 elderly volunteers; 170 mg and 340 mg day^−1^	↑ Wechsler Adult Intelligence Scale, ↓ Pittsburgh Sleep Quality Index; no adverse events or abnormal hematology/biochemistry [101]
Phase I/II completed	Fermented *Camellia* seed peptide-rich extract (Shiseido internal study)	Healthy adults, topical application (open-label)	↑ epidermal CXCL9 expression, proposed to boost immune clearance of senescent fibroblasts [111]
Exploratory cosmetic study	*Camellia*, Tung, and Eucommia peptides	ClinicalTrials.gov and ChiCTR	No interventional trials yet registered

(exegesis: ↑: increase, ↓: decrease).

## Data Availability

No new data were created or analyzed in this study. Data sharing is not applicable to this article.

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
