# Peer review of "Antioxidant Peptides Derived from Woody Oil Resources: Mechanisms of Redox Protection and Emerging Therapeutic Opportunities"

_pharmaceuticals, 2025, doi:10.3390/ph18060842_

Round 1
Reviewer 1 Report
Comments and Suggestions for Authors
Overall, the manuscript is very comprehensive study, but it suffers from several weaknesses. Therefore, it is not recommended for publication at its current state. My comments in detail are as following:
- While the entire manuscript covers research on antioxidant peptides of woody oilseeds, the title does not reflect this.
- The abstract of the review manuscript is written very generally. In this manuscript, which covers a lot of information, the important information obtained within the scope of the study should be highlighted instead of a very generalized abstract.
- Page 2, line 63-64: Author wrote that “Therefore, the high-value processing of by-products from woody oil resources, especially the extraction of active peptides, has attracted widespread attention.” How do we know that this topic is of widespread interest? References should be provided.
- All bacterial names mentioned in the manuscript (Bacillus licheniformis, Bacillus, E. coli, Lactobacillus casei, etc.) should be written in italics. If there is, the culture collection number for the bacteria (such as ATCC) used must be provided.
- In abbreviations, the full name should be written first. For example: Jellyfish collagen Hydrolysate (JCH ) instead of JCH (jellyfish collagen Hydrolysate).
- Figure captions are not sufficient for figures. Although there are explanations in the text, figure captions need to be improved.
- Page5, line 205-210: Author wrote that “In recent years, significant progress has been made in studying the effects of amino acid sequences of antioxidant peptides on oxidative activity. Researchers have identified various amino acids with efficient antioxidant capacity………..” It is not stated what advances have been made in recent years and which researchers have worked on this subject. References should be provided.
- The manuscript includes both gene and protein names. However, it is not specified which of the codes given are genes and which are proteins. According to the rules of gene nomenclature, gene names should be written in italics and protein names should be written normally. The entire manuscript should be reviewed according to this writing rule.
- The manuscript should be enriched with references. Articles such as “Wang, M., et al. (2022). Recent Development in Antioxidant Peptides of Woody Oil Plant By-Products. Food Reviews International, 39(8), 5479–5500. https://doi.org/10.1080/87559129.2022.2073367” can be examined.
Author Response
Comments 1: While the entire manuscript covers research on antioxidant peptides of woody oilseeds, the title does not reflect this.
Response 1: Thank you for pointing this out. We agree with this comment. Therefore, we have revised the title in the manuscript:”Antioxidant peptides derived from woody oil resources:Mechanisms of Redox Protection and Emerging Therapeutic Opportunities”
Comments 2:The abstract of the review manuscript is written very generally. In this manuscript, which covers a lot of information, the important information obtained within the scope of the study should be highlighted instead of a very generalized abstract.
Response 2: Agree. This abstract has been rewritten.”Antioxidant peptides derived from woody oil resources by-products exhibit strong free radi-cal-scavenging abilities and offer potential applications in functional foods, nutraceuticals, and cosmetics. This review summarizes the latest advances in preparation technologies, including enzymatic hydrolysis, microbial fermentation, chemical synthesis, recombinant expression, and molecular imprinting, each with distinct advantages in yield, selectivity, and scalability. The structure–activity relationships of antioxidant peptides are explored with respect to amino acid composition, molecular weight, and 3D conformation, which collectively determine their bioac-tivity and stability. Additionally, emerging delivery systems—such as nanoliposomes, microen-capsulation, and cell-penetrating peptides—are discussed for their role in enhancing peptide sta-bility, absorption, and targeted release. Mechanistic studies reveal that antioxidant peptides from woody oil resources act through network pharmacology, engaging core signaling pathways in-cluding Nrf2/ARE, PI3K/Akt, AMPK, and JAK/STAT, to regulate oxidative stress, mitochondrial health, and inflammation. Preliminary safety data from in vitro, animal, and early clinical studies suggest low toxicity and favorable tolerability. The integration of omics technologies, molecular docking, and bioinformatics is accelerating the mechanism-driven design and functional valida-tion of peptides. In conclusion, antioxidant peptides derived from woody oil resources represent a sustainable, multifunctional, and scalable solution for improving human health and promoting circular bioeconomy. Future research should focus on structural optimization, delivery en-hancement, and clinical validation to facilitate their industrial translation.”
Comments 3: Page 2, line 63-64: Author wrote that “Therefore, the high-value processing of by-products from woody oil resources, especially the extraction of active peptides, has attracted widespread attention.” How do we know that this topic is of widespread interest? References should be provided.
Response 3:Thanks!We have provided the appropriate references. “Therefore, the high-value processing of by-products from woody oil resources, especially the extraction of active peptides, has attracted widespread attention.[20-22].”
References:
- Acevedo‐Juárez, S.; Guajardo‐Flores, D.; Heredia‐Olea, E.; Antunes‐Ricardo, M., Bioactive peptides from nuts: A review. Int. J. Food Sci. Technol. 2022, 57, (4), 2226-2234.
- Boruzi, A. I.; Nour, V., Walnut (Juglans regia L.) leaf powder as a natural antioxidant in cooked pork patties. CyTA - Journal of Food 2019, 17, (1), 431-438.
- Elisha, C.; Bhagwat, P.; Pillai, S., Emerging production techniques and potential health promoting properties of plant and animal protein-derived bioactive peptides. Crit. Rev. Food Sci. Nutr. 2024, 1-30.
Comments 4: All bacterial names mentioned in the manuscript (Bacillus licheniformis, Bacillus, E. coli, Lactobacillus casei, etc.) should be written in italics. If there is, the culture collection number for the bacteria (such as ATCC) used must be provided.
Response 4:Thanks for the comments! This is our negligence. We have revised in the manuscript. “Bacillus licheniformis(Culture No. FCBP‐SB‐0019)(Line 111) 、Lactic Acid Bacteria (LAB)(Line 114) 、 E. coli and S. aureus(Line 337) ”
Comments 5: In abbreviations, the full name should be written first. For example: Jellyfish collagen Hydrolysate (JCH ) instead of JCH (jellyfish collagen Hydrolysate).
Response 5: Thanks for the comments! This is our negligence. We have revised in the manuscript. “Jellyfish collagen Hydrolysate (JCH)”(Line 267)
Comments 6: Figure captions are not sufficient for figures. Although there are explanations in the text, figure captions need to be improved.
Response 6: Thanks for the reviewer’s comments! We have revised in the manuscript.
Figure 1. Schematic route from woody oil resources to antioxidant peptides. Pretreated woody oil seeds or by-products are subjected to either large-scale enzymatic hydrolysis/fermentation or precision synthesis approaches such as chemical or recombinant methods. A subsequent molecular imprinting enrichment step selectively isolates target peptides, which are ultimately applied in food, nutraceutical, and pharmaceutical products.(Line 102-106)
Figure 3. Chemical structures of antioxidant-relevant amino acids (cysteine, histidine, lysine, tryptophan, tyrosine, and glutamic acid), whose thiol, imidazole, ε-amino, indole, phenolic, and γ-carboxylate side chains contribute to radical scavenging and metal-chelating activity in bioactive peptides.(Line 236-239)
Figure 4. Schematic of oral delivery showing gastrointestinal barriers and five carrier strategies: enteric coating, nanoliposomes, nanocarriers, microencapsulation, and cell-penetrating peptides, which safeguard antioxidant peptides en route to systemic circulation and diverse target cells.(Line 305-307)
Figure 5. Antioxidant peptides derived from woody oil resources act via network pharmacology, synchronously engaging Keap1–Nrf2, PI3K–Akt, AMPK, and JAK/STAT pathways to coordi-nate antioxidant gene expression, metabolic-redox homeostasis, and cell survival control. This highlights their multi-target potential for precision nutrition and disease intervention.(Line 494-497)
Comments 7: Page5, line 205-210: Author wrote that “In recent years, significant progress has been made in studying the effects of amino acid sequences of antioxidant peptides on oxidative activity. Researchers have identified various amino acids with efficient antioxidant capacity………..” It is not stated what advances have been made in recent years and which researchers have worked on this subject. References should be provided.
Response 7:Thanks for the comments! This is our negligence. We have revised in the manuscript.“ In recent years, significant progress has been made in studying the effects of amino acid sequences of antioxidant peptides on oxidative activity[48]. Researchers have identified various amino acids with efficient antioxidant capacity[49], such as glutamic acid, cysteine[50], histidine, lysine, tryptophan, and tyrosine[51].”
References:
- Zou, T.-B.; He, T.-P.; Li, H.-B.; Tang, H.-W.; Xia, E.-Q., The Structure-Activity Relationship of the Antioxidant Peptides from Natural Proteins. Molecules 2016, 21, (1).
- Yang, X.-R.; Zhang, L.; Ding, D.-G.; Chi, C.-F.; Wang, B.; Huo, J.-C., Preparation, Identification, and Activity Evaluation of Eight Antioxidant Peptides from Protein Hydrolysate of Hairtail (Trichiurus japonicas) Muscle. Marine Drugs 2019, 17, (1).
- Lopez-Huertas, E.; Alcaide-Hidalgo, J. M., Characterisation of Endogenous Peptides Present in Virgin Olive Oil. Int. J. Mol. Sci. 2022, 23, (3).
- Shi, C.; Liu, M.; Zhao, H.; Lv, Z.; Liang, L.; Zhang, B., A Novel Insight into Screening for Antioxidant Peptides from Hazelnut Protein: Based on the Properties of Amino Acid Residues. Antioxidants 2022, 11, (1).
Comments 8: The manuscript includes both gene and protein names. However, it is not specified which of the codes given are genes and which are proteins. According to the rules of gene nomenclature, gene names should be written in italics and protein names should be written normally. The entire manuscript should be reviewed according to this writing rule.
Response 8:Thanks for the comments! This is our negligence. We have revised in the manuscript.“CASP12(Line 554)、miR-155(Line 553)、miR-27a(Line 556)、IL-1β(Line 613)、LKB1(Line 614)”
Comments 9: The manuscript should be enriched with references. Articles such as “Wang, M., et al. (2022). Recent Development in Antioxidant Peptides of Woody Oil Plant By-Products. Food Reviews International, 39(8), 5479–5500. https://doi.org/10.1080/87559129.2022.2073367” can be examined.
Response 9:Thanks for the comments! We have revised in the manuscript.
Producing antioxidant peptides from woody oil resources not only enhances the added value of the oil but also increases its potential applications in food, cosmetics, and health products, while helping to promote the effective use of agricultural by-products and supporting sustainable agricultural practices[17].
Reference:
- Wang, M.; Wu, W.; Xiao, J.; Li, C.; Chen, B.; Shen, Y., Recent Development in Antioxidant Peptides of Woody Oil Plant By-Products. Food Reviews International 2022, 39, (8), 5479-5500.

Reviewer 2 Report
Comments and Suggestions for Authors
The manuscript presents a comprehensive and timely review of antioxidant peptides derived from woody oilseed proteins. The manuscript effectively highlights underexplored protein resources from woody oilseeds and their therapeutic potential, especially in the context of oxidative stress-related diseases. It is well-structured, detailed, and highly relevant to the fields of functional foods, nutraceuticals, and biopharmaceuticals. The manuscript can be accepted for publication after some minor changes.
Comments:
- Include more descriptive captions for figures.
- Include a comparative summary table listing advantages/disadvantages of microbial fermentation, chemical hydrolysis, enzymatic methods, recombinant DNA technology, CAD, and MIPs.
- Include a figure explaining structure-activity relationship of antioxidant peptides.
- Include a separate section on safety, immunogenicity, or toxicity of these peptides.
- Discuss clinical translation or ongoing clinical trials involving antioxidant peptides from woody oilseeds.
Author Response
Comments 1: Include more descriptive captions for figures.
Response 1: Thank you for pointing this out. We agree with this comment. Therefore, we have revised the title in the manuscript.
Figure 1. Schematic route from woody oil resources to antioxidant peptides. Pretreated woody oil seeds or by-products are subjected to either large-scale enzymatic hydrolysis/fermentation or precision synthesis approaches such as chemical or recombinant methods. A subsequent molecular imprinting enrichment step selectively isolates target peptides, which are ultimately applied in food, nutraceutical, and pharmaceutical products.(Line 102-106)
Figure 3. Chemical structures of antioxidant-relevant amino acids (cysteine, histidine, lysine, tryptophan, tyrosine, and glutamic acid), whose thiol, imidazole, ε-amino, indole, phenolic, and γ-carboxylate side chains contribute to radical scavenging and metal-chelating activity in bioactive peptides.(Line 236-239)
Figure 4. Schematic of oral delivery showing gastrointestinal barriers and five carrier strategies: enteric coating, nanoliposomes, nanocarriers, microencapsulation, and cell-penetrating peptides, which safeguard antioxidant peptides en route to systemic circulation and diverse target cells.(Line 305-307)
Figure 5. Antioxidant peptides derived from woody oil resources act via network pharmacology, synchronously engaging Keap1–Nrf2, PI3K–Akt, AMPK, and JAK/STAT pathways to coordi-nate antioxidant gene expression, metabolic-redox homeostasis, and cell survival control. This highlights their multi-target potential for precision nutrition and disease intervention.(Line 494-497)
Comments 2:Include a comparative summary table listing advantages/disadvantages of microbial fermentation, chemical hydrolysis, enzymatic methods, recombinant DNA technology, CAD, and MIPs.
Response 2: Thanks for the comments! This is our negligence. We have added table 2 to the manuscript.
Table 2. Advantages and Limitation of different preparation methods for antioxidant peptides
Comments 3: Include a figure explaining structure-activity relationship of antioxidant peptides.
Response 3:Thanks!We have added Figure 2 to the manuscript.
Figure 2. The antioxidant potency of bioactive peptides is dictated by three interlinked factors: amino acid composition, molecular weight range, and three-dimensional conformation.
Comments 4: Include a separate section on safety, immunogenicity, or toxicity of these peptides.
Response 4:Thanks for the comments! We have revised in the manuscript.
“6. Safety, Immunogenicity and Toxicity of Antioxidant Peptides Derived from Woody Oil Resources”(From Line 572-592)
Table 3. Safety modeling of antioxidant peptides from woody oil resources
Comments 5: Discuss clinical translation or ongoing clinical trials involving antioxidant peptides from woody oilseeds.
Response 5:Thanks for the comments! This is our negligence. We have revised in the manuscript.
“7. Clinical Translation and Ongoing Human Studies”(Line 593-595)
Table 4. Clinical translation of antioxidant peptides derived from woody oil resources

Reviewer 3 Report
Comments and Suggestions for Authors
The manuscript entitled: Exploring the preparation, utilization and therapeutic potential of woody oilseed protein resources: a review presents an interesting and useful overview of the currently known efficacy and utilization of woody oilseed protein resources in various fields.
The manuscript is well written and documented and provides the latest findings regarding the mechanism and utilization of woody oilseed proteins. The references are sufficient and meaningful for the topic covered.
However, the main problem of the manuscript is the English language, which is accurate enough, but the sentences used are too long and unclear. This leads to a text that is difficult to understand (some examples: lines 74-79; 79-83; 123-133; 141-145; 169-174; 180-185). The Abstract also has the same problem, which makes it difficult to understand. Please revise the text while retaining the basic idea, but with shorter sentences.
Also change the expression: In his article, Ishak, with In their article, Ishak and Sarbon.
Section 4's title (Absorption and transport of antioxidant peptides) is misleading and it is recommended to change it to: Delivery systems with antioxidant peptides.
For Section 5 (Studies on the signaling pathway regulation mechanism of antioxidant peptides from woody oilseed sources) it is recommended to either rename Section 5.3. to 6. or rename the title of Section 5 to: Studies on the mechanism of signaling pathway regulation and potential use of antioxidant peptides from woody oilseed sources
Comments on the Quality of English Language
The English language must be revised as recommendations
Author Response
Comments 1: However, the main problem of the manuscript is the English language, which is accurate enough, but the sentences used are too long and unclear. This leads to a text that is difficult to understand (some examples: lines 74-79; 79-83; 123-133; 141-145; 169-174; 180-185). The Abstract also has the same problem, which makes it difficult to understand. Please revise the text while retaining the basic idea, but with shorter sentences.
Response 1: Thanks for the comments! This is our negligence. We have revised in the manuscript.
Line 74-83 has been modified to“Antioxidant peptides can be prepared using various methods, with the appropriate technique selected according to specific needs (Figure 1). Peptides with antioxidant activity can be efficiently released from natural proteins through enzymatic hydrolysis and microbial fermentation, both of which offer mild reaction conditions and high se-lectivity, making them suitable for use in functional foods, nutraceuticals, and phar-maceuticals. Although chemical hydrolysis is less selective, it is cost-effective, easy to operate, and suitable for large-scale production, and has thus been widely applied in the food, feed, and cosmetic industries.”(Line 85-92)
Line 123-133 has been modified to“Wang et al.[26] hydrolyzed walnut proteins using trypsin and a combination of en-zymes, and found that the resulting walnut peptides could improve (Lipopolysaccha-rides)LPS-induced memory deficits by modulating inflammatory responses and oxi-dative stress in the brain. In addition, Liu et al.[27] used neutral enzymes to hydrolyze walnut proteins and obtained peptides with high (Angiotensin Converting En-zyme)ACE inhibitory activity, suggesting their potential as functional food ingredients with anti-hypertensive properties.Overall, the widespread application of these three methods has greatly contrib-uted to the rapid development of peptide-based products across various industries. Each method offers distinct advantages and can complement the others to meet diverse needs and technical requirements.”(Line 133-144)
Line 141-145 has been modified to“Chemical synthesis is a method used to produce peptides with specific amino acid sequences through chemical reactions. It allows precise control over the sequence and is widely applied in drug discovery, functional food development, and vaccine research. This approach is particularly well-suited for the rapid synthesis of short- and medi-um-length peptides.”(Line 146-150)
Line 169-174 has been modified to“The core advantage of molecularly imprinted polymer (MIP) post-modification technology lies in its ability to mimic natural antibodies by creating polymers with specific recognition sites that can accurately identify and bind target peptides or pro-teins[35]. ”(Line 182-184)
Line 180-185 has been modified to“Due to their high specificity, MIPs are often used to extract target bioactive pep-tides from complex biological samples. For example, MIPs can selectively capture spe-cific peptides during protein extraction, reducing reliance on large volumes of chemical reagents commonly used in traditional purification methods, thereby minimizing en-vironmental impact and lowering costs [37].”(Line 194-198)
Comments 2:Also change the expression: In his article, Ishak, with In their article, Ishak and Sarbon.
Response 2: Thanks for the comments! This is our negligence. We have revised in the manuscript.
“In their article, Ishak and Sarbon [53] mentions that...”(Line 243)
Comments3:Section4's title (Absorption and transport of antioxidant peptides) is misleading and it is recommended to change it to: Delivery systems with antioxidant peptides.
Response 3:Thanks for the comments!We have revised in the manuscript.
- Delivery systems with antioxidant peptides(Line 294)
Comments4:For Section 5 (Studies on the signaling pathway regulation mechanism of antioxidant peptides from woody oilseed sources) it is recommended to either rename Section 5.3. to 6. or rename the title of Section 5 to: Studies on the mechanism of signaling pathway regulation and potential use of antioxidant peptides from woody oilseed sources.
Response 4:Thanks for the comments! We have revised in the manuscript.
“8. Application Prospects and Industrialization Potential
Pathway-specific antioxidant peptides derived from antioxidant peptides derived from woody oil resources demonstrate promising applications across health-related industries. By selectively activating key signaling pathways such as Nrf2/ARE, PI3K/Akt, and AMPK, these peptides enhance cellular antioxidant defenses and sup-press chronic inflammation, especially relevant for aging populations and those with metabolic disorders.
(1) Precision Nutrition for Aging and Metabolic Syndromes
Food-derived peptides[111] may have greater stability, safety, absorption effi-ciency and biological activity, properties that give them greater potential to alleviate diseases associated with oxidative damage than other antioxidant drugs or phyto-chemicals. Tonolo et al.[112] synthesized four peptides from milk protein sequences and confirmed their antioxidant effects through Nrf2 nuclear translocation in Caco-2 cells. Among them, KVLPVPEK showed the strongest activation of antioxidant gene expression via the Keap1-Nrf2 pathway.
Yang et al.[113] demonstrated that BSP1 (KKWNP), a peptide from black soybean, reduced lead-induced neuroinflammation by inhibiting NF-κB phosphorylation and suppressing inflammatory cytokines such as IL-1β, TNF-α, NLRP3, and IL-18. Notably, BSP1 also activated the AMPK/SIRT1 pathway by promoting LKB1 and AMPK phos-phorylation, offering neuroprotective potential against Alzheimer’s disease.
For metabolic diseases like type 2 diabetes, peptides with α-glucosidase inhibition or glycogen synthesis regulation are being explored.[114]. Jiang et al.[115] showed that Vglycin, a 37-residue peptide isolated from pea seeds, improved insulin sensitivity via PI3K/Akt-mediated phosphorylation of GSK3α/β and upregulation of GLUT4 expres-sion, suggesting its potential in glycemic control.
(2) Functional Foods via Multi-Pathway Synergy
To amplify biological effects, antioxidant peptides can be complexed with phe-nolic compounds, forming peptide–phenol conjugates that exploit synergistic mul-ti-pathway activation. Choi et al. [116] demonstrated that coupling the oyster peptide QHGV with phenolic acids such as caffeic, gallic, and ferulic acids enhanced its anti-oxidant activity and suppressed LPS-induced inflammation in macrophages. These conjugates inhibited ROS production and the expression of iNOS and COX-2, while downregulating the phosphorylation of JNK, ERK, and p38 MAPK, highlighting their potential as next-generation functional food ingredients.
(3) Cosmeceutical Applications: Anti-UV and Skin Barrier Repair
Topical application of antioxidant peptides holds potential in anti-photoaging skincare. Cheng et al.[117] found that three yeast-derived fermentation actives could alleviate H₂O₂ and UVA-induced oxidative damage in human fibroblasts by activating the PI3K/Akt pathway.
In parallel, Song et al.[118] showed that fermented oat bran extract downregu-lated JAK/STAT signaling, suppressed inflammatory cytokine transcription, and im-proved skin barrier function in UVB-damaged keratinocytes. These results support the development of anti-UV serums and barrier-repairing formulations based on antioxi-dant peptides.
(4) Mechanism-Guided Peptide Innovation for Industrial Translation
With advancements in omics technologies, molecular docking, and path-way-targeted validation, peptide development is shifting from empirical screening to mechanism-based innovation. This transition supports the industrialization of antiox-idant peptides derived from woody oil resources antioxidant peptides as high-value bioactives in nutrition, functional foods, and cosmeceuticals.”( From Line 597-645)

Reviewer 4 Report
Comments and Suggestions for Authors
Certain corrections would significantly improve this paper after which it could be accepted for publication. Suggestions are listed below.
1) Some of the claims through the text are presented as general conclusions, but they should be supported by the addition of appropriate references. Lines 103 – 110, description of a chemical hydrolysis method should be supported by references and some examples, if available. Additionally, (subsection 2.2) the explanation of the significance of computer-driven technology; “These amino acids exhibit antioxidant effects due to the chemical nature of their side chains, which can directly react with free radicals.” (subsection 3.1.); (subsection 3.2.) “The optimal molecular weight range…”…
2) Some abbreviations need to be explained, Line 130 (GLP and MGLP), Line 218 (DPPH), Line 219 (MCP), Line 254 (PEF), Line 286 (SIF, SGF), Line 360 (AAV)…
3) Line 201, if there are multiple optimization methods, they should be listed here.
4) The text does not state that Figure 4 is being commented on.
5) Subsections 5.1. and 5.2. are too extensive, and it would be useful to focus on the significance of the potential effect of oilseed proteins on these signaling pathways.
6) The authors are encouraged to add more keywords.
7) Consider an addition of Future Perspectives.
Author Response
Response to Reviewer 4 Comments
Comments 1: Some of the claims through the text are presented as general conclusions, but they should be supported by the addition of appropriate references. Lines 103 – 110, description of a chemical hydrolysis method should be supported by references and some examples, if available. Additionally, (subsection 2.2) the explanation of the significance of computer-driven technology; “These amino acids exhibit antioxidant effects due to the chemical nature of their side chains, which can directly react with free radicals.” (subsection 3.1.); (subsection 3.2.) “The optimal molecular weight range…”
Response 1: Thanks for the comments! This is our negligence. We have revised in the manuscript.
Line 103-110 has been modified to“Chemical hydrolysis is a method of obtaining small molecule peptides by treating proteins with acid or alkali solutions to break their peptide bonds[25].”(Line 121-122)
Line 151-154 has been modified to“Computer-driven technology (CAD)[31], on the other hand, optimizes peptide sequence design, three-dimensional structure prediction and bioactivity assessment through computer-aided design[32], bioinformatics analysis and artificial intelligence technology, greatly improving the efficiency and precision of peptide design.”(Line 166-169)
Line 214-216 has been modified to“These amino acids exhibit antioxidant effects due to the chemical nature of their side chains, which can directly react with free radicals[52]. ”(Line 234-236)
Line 238-241 has been modified to“The optimal molecular weight range for antioxidant peptides is not well defined, but it is generally believed that smaller peptides are more likely to penetrate cell membranes and function inside the cell[56, 57], interacting with target free radicals and thus exerting antioxidant effects. ”(Line 262-265)
References:
- Khiari, Z.; Mason, B., Comparative dynamics of fish by-catch hydrolysis through chemical and microbial methods. Lwt 2018, 97, 135-143.
- Hudon, A.; Gaudreau-Ménard, C.; Bouchard-Boivin, M.; Godin, F.; Cailhol, L., The Use of Computer-Driven Technologies in the Treatment of Borderline Personality Disorder: A Systematic Review. J. Clin. Med. 2022, 11, (13).
- Malibari, A. A.; Alzahrani, J. S.; Eltahir, M. M.; Malik, V.; Obayya, M.; Duhayyim, M. A.; Lira Neto, A. V.; de Albuquerque, V. H. C., Optimal deep neural network-driven computer aided diagnosis model for skin cancer. Comput. Electr. Eng. 2022, 103.
- Matsui, R.; Honda, R.; Kanome, M.; Hagiwara, A.; Matsuda, Y.; Togitani, T.; Ikemoto, N.; Terashima, M., Designing antioxidant peptides based on the antioxidant properties of the amino acid side-chains. Food Chem. 2018, 245, 750-755.
- Landi, N.; Clemente, A.; Pedone, P. V.; Ragucci, S.; Di Maro, A., An Updated Review of Bioactive Peptides from Mushrooms in a Well-Defined Molecular Weight Range. Toxins 2022, 14, (2).
- Wang, J.; Wang, K.; Lin, S.; Zhao, P.; Jones, G.; Trang, H.; Liu, J.; Ye, H., Improvement of antioxidant activity of peptides with molecular weights ranging from 1 to 10kDa by PEF technology. Int. J. Biol. Macromol. 2012, 51, (3), 244-249.
Comments 2:Some abbreviations need to be explained, Line 130 (GLP and MGLP), Line 218 (DPPH), Line 219 (MCP), Line 254 (PEF), Line 286 (SIF, SGF), Line 360 (AAV)…
Response 2: Thanks for the comments! This is our negligence. We have revised in the manuscript.“human glucagon-like peptide-1 (MGLP-1)(Line 151)、(Glucagon-like peptide-1)GLP-1(Line 152)、2,2-Diphenyl-1-picrylhydrazyl (DPPH)(Line 248)、Jellyfish collagen Hydrolysate (JCH)(Line 269)、marine collagen peptide (MCP)(Line 250)、pulsed electric fields (PEF)(Line285) 、nanoparticles (NPs)(Line 313)、chitosan/alginate nanoparticles (CS-TPP-ALG)(Line 319)、simulated intestinal fluid (SIF)(Line 319)、simulated gastric fluid (SGF)(Line 320)、poly lactic-co-glycolic acid (PLGA)(Line 360) 、adeno-associated virus (AAV)(Line 395)”
Comments3:Line 201, if there are multiple optimization methods, they should be listed here.
Response 3:Thanks for the comments!We believe that 3.1-3.3 illustrate the optimization approach.(Line 230-295)
3.1. Effects of amino acid sequence on activity
3.2. Effects of molecular weight on activity
3.3. Effects of three-dimensional structure on activity
Comments4:The text does not state that Figure 4 is being commented on.
Response 4:Thanks for the comments! This is our negligence.We have revised in the manuscript.
“Antioxidant peptides derived from woody oil resources woody oil resources ex-hibit unique network effects in the regulation of biological activities that do not depend on the linear transmission of a single signaling pathway (Figure 5)...”(Line 464-466)
Comments 5: Subsections 5.1. and 5.2. are too extensive, and it would be useful to focus on the significance of the potential effect of oilseed proteins on these signaling pathways.
Response 5:Thanks for the comments! We have revised in the manuscript.
“5.1. Integrated regulatory mechanisms of core antioxidant signaling pathways
5.1.1. Keap1/Nrf2/ARE pathway: A Redox-Sensing Hub for Peptide Intervention
The Keap1/Nrf2/ARE pathway is a well-characterized oxidative stress response mechanism that is highly responsive to antioxidant peptides derived from antioxidant peptides derived from woody oil resources. These peptides can bind to Keap1, releas-ing its inhibitory effect on Nrf2, which then translocates to the nucleus and activates ARE elements to promote the expression of antioxidant enzymes such as HO-1 and NQO1.
For instance, Zhong et al.[29] identified four functional peptides (e.g., PCRGVLLR, KVLPVPQKA) from walnut proteins fermented by Lactobacillus casei, which showed strong binding affinities to Keap1 (−7.4 to −8.3 kcal/mol) and significantly enhanced antioxidant capacity in vivo. Similarly, Qi et al.[88] reported that the walnut-derived peptide LPLLR inhibited Keap1–Nrf2 interaction through steric hindrance, resulting in increased SOD and CAT activity and decreased MDA levels in H₂O₂-exposed Caco-2 cells, indicating its structural targeting potential.
5.1.2. PI3K/Akt pathway: Enhancing Redox Signaling and Cytoprotection
The PI3K/Akt pathway is a key intracellular signaling cascade that bridges oxida-tive stress regulation and cell survival. It has been shown that Antioxidant peptides derived from woody oil resources can activate this pathway to indirectly enhance Nrf2-mediated transcription of antioxidant genes.
Several studies demonstrated that walnut-derived peptides restore PI3K/Akt sig-naling under oxidative damage conditions (e.g., H2O2 exposure), inhibit apopto-sis-related proteins, and promote antioxidant enzyme activity[89, 90].
5.1.3. JAK/STAT pathway: Coordinating Anti-Inflammatory and Redox Responses
The JAK/STAT pathway plays a dual role in immune regulation and oxidative stress defense. Recent studies have shown that certain antioxidant peptides derived from woody oil resources can inhibit the IL-6/JAK2/STAT3 axis to reduce neuroin-flammation and oxidative damage.
For example, Zhang et al. [91] found that the heptapeptide WCPFSRSF attenuated cognitive impairment induced by sleep deprivation, partially by inhibiting microglial activation and suppressing the JAK/STAT signaling pathway. Additional studies using in vitro models have demonstrated that these peptides can lower ROS levels, upregu-late endogenous antioxidant enzymes, and downregulate pro-inflammatory cytokines, achieving a dual anti-inflammatory and antioxidant effect.
5.2. Exploration of auxiliary pathways and systemic crossover mechanisms
5.2.1. AMPK pathway: Energy-Redox Coupling and Mitochondrial Repair
AMPK (AMP-activated protein kinase) serves as a central regulator of cellular responses to energy stress (e.g., elevated AMP/ATP ratio). Upon activation, it inhibits mTOR signaling, initiates autophagy, and promotes mitochondrial biogenesis via the SIRT1/PGC-1α axis—thereby restoring both energy and redox balance[92].
Recent studies have highlighted AMPK as a critical node in the action of various antioxidant peptides. For instance, the dietary peptide QEPV was shown to enhance antioxidant enzyme activity and suppress lipid peroxidation in RAW 264.7 cells via activation of the AMPK and PPARα pathways[93]. Similarly, the peptide DDWEN-WAK from perch hydrolysate improved mitochondrial membrane potential, structure, and ATP production in H2O2 -stressed Caco-2 cells by activating both the AMPK and Nrf2 pathways [94]. These findings support the potential of natural protein-derived peptides to modulate AMPK signaling for energy metabolism and oxidative stress protection.
If antioxidant peptides derived from woody oil resources can be further validated to activate AMPK and enhance mitochondrial repair, they could serve as promising agents for countering metabolic stress in functional food applications.
5.2.2. Emerging mechanisms: non-coding RNAs and microecological interactions for regulation
Beyond classical signaling pathways, non-coding RNAs (e.g., microRNAs) and gut microbiota have emerged as key players in shaping the bioactivity of antioxidant pep-tides, forming a complex peptide–epigenetic–microecological regulatory axis. Yan et al.[95]reported that the antimicrobial peptide YD downregulated miR-155 expression, which relieved suppression of CASP12 and inhibited the NF-κB pathway—ultimately reducing inflammation and oxidative stress. This indicates that peptides can modulate miRNA networks to control immune and oxidative responses. Similarly, miR-27a has been shown to regulate the PPARγ–PI3K/Akt–GLUT4 axis, thereby improving in-sulin resistance in obese mice and 3T3-L1 adipocytes[96].
Peptides influencing miRNA expression could therefore fine-tune cellular metab-olism and cytoprotection.
In parallel, peptide–microbiota interactions are gaining attention. High collagen peptide intake has been shown to reshape gut microbial composition and increase the production of short-chain fatty acids (SCFAs)[97] SCFAs such as acetate, propionate, and butyrate can activate AMPK/PPARγ signaling and suppress c-Jun phosphoryla-tion, thereby inhibiting hepatic stellate cell (LX2) activation and exerting anti-fibrotic effects [98]
While current studies have not directly demonstrated the involvement of antiox-idant peptides derived from woody oil resources in miRNA or microbiota regulation, their unique amino acid profiles suggest strong potential in these emerging mecha-nisms. Future research incorporating the gut–brain axis, miRNA expression profiling, and metabolomics could further reveal their systemic antioxidant and an-ti-inflammatory capabilities.”( Line 500-572)
Comments 6:The authors are encouraged to add more keywords.
Response 6:Thanks for the comments! We have revised in the manuscript.
“Keywords:Antioxidant peptides; Woody Oil Resources; Signal transduction pathways; Keap1/Nrf2/ARE; PI3K/Akt; AMPK; JAK/STAT; Bioavailability; Functional foods; Molecular docking”(Line 36-38)
Comments 7:Consider an addition of Future Perspectives.
Response 7:Thanks for the comments! This is our negligence.We have revised in the manuscript.
“9. Future Perspectives
Antioxidant peptides derived from antioxidant peptides derived from woody oil resources (e.g., Camellia, walnut) show strong promise as natural agents for oxidative stress management, but several challenges remain before large-scale application can be achieved.
(1) Structure–function elucidation
Future studies should focus on clarifying the structure–activity relationship (SAR) of key peptide sequences using tools like molecular docking, quantitative struc-ture-activity relationship (QSAR) modeling, and AI-driven prediction[106].
(2) Improved bioavailability
Developing effective delivery systems (e.g., nanoemulsions, enteric coatings) is essential to protect peptides from gastrointestinal degradation and enhance systemic absorption[119].
(3) Mechanistic integration via multi-omics.
Combining transcriptomics, proteomics, and microbiomics will help unravel how these peptides influence redox pathways, gut–brain signaling, and host metabo-lism[120].
(4) Human validation
Though walnut peptides have entered clinical trials[121], more human studies are needed for other woody oil resources to assess safety, efficacy, and long-term effects.
(5) Sustainable industrialization
Utilizing seed cake proteins from oil production to generate high-value peptides supports circular bioeconomy goals and clean-label product development[122].”(Line 646-666)

Round 2
Reviewer 1 Report
Comments and Suggestions for Authors
- I am pleased to inform that the authors have addressed all the reviewer's comments and have considerably revised their manuscript. So, I recommend to accept this manuscript.

Reviewer 3 Report
Comments and Suggestions for Authors
Thank you for the replies